# A Multi-World Approach to Question Answering about Real-World Scenes based on Uncertain Input

**Mateusz Malinowski**       **Mario Fritz**
Max Planck Institute for Informatics
Saarbrücken, Germany
`{mmalinow,mfritz}@mpi-inf.mpg.de`

## Abstract

We propose a method for automatically answering questions about images by bringing together recent advances from natural language processing and computer vision. We combine discrete reasoning with uncertain predictions by a multi-world approach that represents uncertainty about the perceived world in a bayesian framework. Our approach can handle human questions of high complexity about realistic scenes and replies with range of answer like counts, object classes, instances and lists of them. The system is directly trained from question-answer pairs. We establish a first benchmark for this task that can be seen as a modern attempt at a visual turing test.

## 1   Introduction

As vision techniques like segmentation and object recognition begin to mature, there has been an increasing interest in broadening the scope of research to full scene understanding. But what is meant by "understanding" of a scene and how do we measure the degree of "understanding"? Most often "understanding" refers to a correct labeling of pixels, regions or bounding boxes in terms of semantic annotations. All predictions made by such methods inevitably come with uncertainties attached due to limitations in features or data or even inherent ambiguity of the visual input.

Equally strong progress has been made on the language side, where methods have been proposed that can learn to answer questions solely from question-answer pairs [1]. These methods operate on a set of facts given to the system, which is refered to as a world. Based on that knowledge the answer is inferred by marginalizing over multiple interpretations of the question. However, the correctness of the facts is a core assumption.

We like to unite those two research directions by addressing a question answering task based on real-world images. To combine the probabilistic output of state-of-the-art scene segmentation algorithms, we propose a Bayesian formulation that marginalizes over multiple possible worlds that correspond to different interpretations of the scene.

To date, we are lacking a substantial dataset that serves as a benchmark for question answering on real-world images. Such a test has high demands on "understanding" the visual input and tests a whole chain of perception, language understanding and deduction. This very much relates to the "AI-dream" of building a turing test for vision. While we are still not ready to test our vision system on completely unconstrained settings that were envisioned in early days of AI, we argue that a question-answering task on complex indoor scenes is a timely step in this direction.

**Contributions:**   In this paper we combine automatic, semantic segmentations of real-world scenes with symbolic reasoning about questions in a Bayesian framework by proposing a multi-world approach for automatic question answering. We introduce a novel dataset of more than 12,000

question-answer pairs on RGBD images produced by humans, as a modern approach to a visual turing test. We benchmark our approach on this new challenge and show the advantages of our multi-world approach. Furthermore, we provide additional insights regarding the challenges that lie ahead of us by factoring out sources of error from different components.

## 2 Related work

**Semantic parsers**: Our work is mainly inspired by [1] that learns the semantic representation for the question answering task solely based on questions and answers in natural language. Although the architecture learns the mapping from weak supervision, it achieves comparable results to the semantic parsers that rely on manual annotations of logical forms ([2], [3]). In contrast to our work, [1] has never used the semantic parser to connect the natural language to the perceived world.

**Language and perception**: Previous work [4, 5] has proposed models for the language grounding problem with the goal of connecting the meaning of the natural language sentences to a perceived world. Both methods use images as the representation of the physical world, but concentrate rather on constrained domain with images consisting of very few objects. For instance [5] considers only two mugs, monitor and table in their dataset, whereas [4] examines objects such as blocks, plastic food, and building bricks. In contrast, our work focuses on a diverse collection of real-world indoor RGBD images [6] - with many more objects in the scene and more complex spatial relationship between them. Moreover, our paper considers complex questions - beyond the scope of [4] and [5] - and reasoning across different images using only textual question-answer pairs for training. This imposes additional challenges for the question-answering engines such as scalability of the semantic parser, good scene representation, dealing with uncertainty in the language and perception, efficient inference and spatial reasoning. Although others [7, 8] propose interesting alternatives for learning the language binding, it is unclear if such approaches can be used to provide answers on questions.

**Integrated systems that execute commands**: Others [9, 10, 11, 12, 13] focus on the task of learning the representation of natural language in the restricted setting of executing commands. In such scenario, the integrated systems execute commands given natural language input with the goal of using them in navigation. In our work, we aim for less restrictive scenario with the question-answering system in the mind. For instance, the user may ask our architecture about counting and colors ('How many green tables are in the image?'), negations ('Which images do not have tables?') and superlatives ('What is the largest object in the image?').

**Probabilistic databases**: Similarly to [14] that reduces Named Entity Recognition problem into the inference problem from probabilistic database, we sample multiple-worlds based on the uncertainty introduced by the semantic segmentation algorithm that we apply to the visual input.

## 3 Method

Our method answers on questions based on images by combining natural language input with output from visual scene analysis in a probabilistic framework as illustrated in Figure 1. In the single world approach, we generate a single perceived world $\mathcal{W}$ based on segmentations - a unique interpretation of a visual scene. In contrast, our multi-world approach integrates over many latent worlds $\mathcal{W}$, and hence taking different interpretations of the scene and question into account.

**Single-world approach for question answering problem** We build on recent progress on end-to-end question answering systems that are solely trained on question-answer pairs $(Q, A)$ [1]. Top part of Figure 1 outlines how we build on [1] by modeling the logical forms associated with a question as latent variable $\mathcal{T}$ given a single world $\mathcal{W}$. More formally the task of predicting an answer $\mathcal{A}$ given a question $\mathcal{Q}$ and a world $\mathcal{W}$ is performed by computing the following posterior which marginalizes over the latent logical forms (semantic trees in [1]) $\mathcal{T}$:

$$P(A|Q, \mathcal{W}) := \sum_{\mathcal{T}} P(A|\mathcal{T}, \mathcal{W}) P(\mathcal{T}|Q). \tag{1}$$

$P(A|\mathcal{T}, \mathcal{W})$ corresponds to denotation of a logical form $\mathcal{T}$ on the world $\mathcal{W}$. In this setting, the answer is unique given the logical form and the world: $P(A|\mathcal{T}, \mathcal{W}) = \mathbf{1}[A \in \sigma_{\mathcal{W}}(\mathcal{T})]$ with the evaluation function $\sigma_{\mathcal{W}}$, which evaluates a logical form on the world $\mathcal{W}$. Following [1] we use DCS Trees that yield the following recursive evaluation function $\sigma_{\mathcal{W}}$: $\sigma_{\mathcal{W}}(\mathcal{T}) :=$

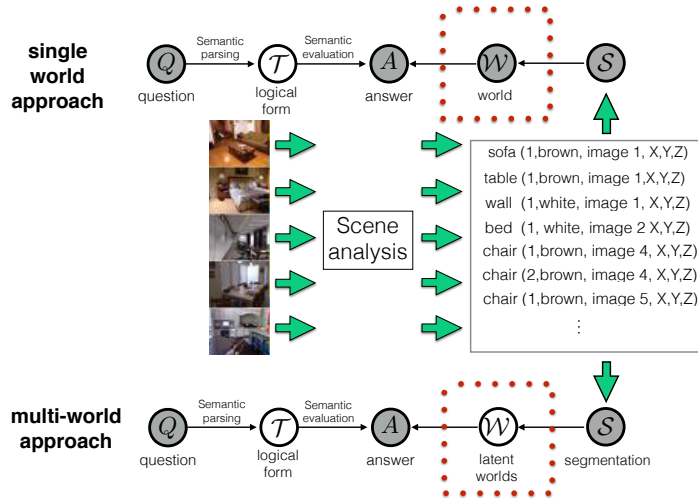

Figure 1: Overview of our approach to question answering with multiple latent worlds in contrast to single world approach.

$\bigcap_j^d \{v \ : \ v \in \sigma_{\mathcal{W}}(p), \ t \in \sigma_{\mathcal{W}}(\mathcal{T}_j), \ \mathcal{R}_j(v,t)\}$ where $\mathcal{T} := \langle p, (\mathcal{T}_1, \mathcal{R}_1), (\mathcal{T}_2, \mathcal{R}_2), ..., (\mathcal{T}_d, \mathcal{R}_d)\rangle$ is the semantic tree with a predicate $p$ associated with the current node, its subtrees $\mathcal{T}_1, \mathcal{T}_2, ..., \mathcal{T}_d$, and relations $\mathcal{R}_j$ that define the relationship between the current node and a subtree $\mathcal{T}_j$.

In the predictions, we use a log-linear distribution $P(\mathcal{T}|Q) \propto \exp(\theta^T \phi(Q, \mathcal{T}))$ over the logical forms with a feature vector $\phi$ measuring compatibility between $Q$ and $\mathcal{T}$ and parameters $\theta$ learnt from training data. Every component $\phi_j$ is the number of times that a specific feature template occurs in $(Q, \mathcal{T})$. We use the same templates as [1]: string triggers a predicate, string is under a relation, string is under a trace predicate, two predicates are linked via relation and a predicate has a child. The model learns by alternating between searching over a restricted space of valid trees and gradient descent updates of the model parameters $\theta$. We use the Datalog inference engine to produce the answers from the latent logical forms. The linguistic phenomena such as superlatives and negations are handled by the logical forms and the inference engine. For a detailed exposition, we refer the reader to [1].

**Question answering on real-world images based on a perceived world** Similar to [5], we extend the work of [1] to operate now on what we call *perceived world* $\mathcal{W}$. This still corresponds to the single world approach in our overview Figure 1. However our world is now populated with "facts" derived from automatic, semantic image segmentations $\mathcal{S}$. For this purpose, we build the world by running a state-of-the-art semantic segmentation algorithm [15] over the images and collect the recognized information about objects such as object class, 3D position, and

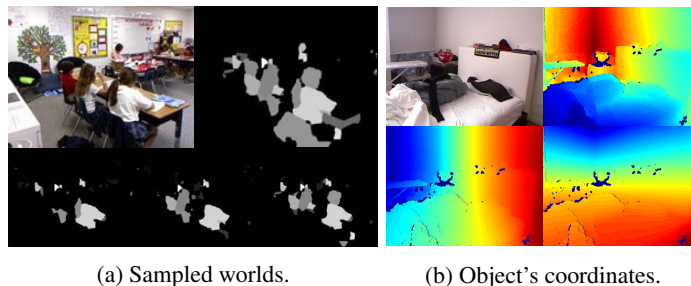

(a) Sampled worlds.                    (b) Object's coordinates.

Figure 2: Fig. 2a shows a few sampled worlds where only segments of the class 'person' are shown. In the clock-wise order: original picture, most confident world, and three possible worlds (gray-scale values denote the class confidence). Although, at first glance the most confident world seems to be a reasonable approach, our experiments show opposite - we can benefit from imperfect but multiple worlds. Fig. 2b shows object's coordinates (original and $Z$, $Y$, $X$ images in the clock-wise order), which better represent the spatial location of the objects than the image coordinates.

|  | Predicate | Definition |
|---|---|---|
| auxiliary relations | $closeAbove(A, B)$ | $above(A, B)$ and $(Y_{min}(B) < Y_{max}(A) + \epsilon)$ |
|  | $closeLeftOf(A, B)$ | $leftOf(A, B)$ and $(X_{min}(B) < X_{max}(A) + \epsilon)$ |
|  | $closeInFrontOf(A, B)$ | $inFrontOf(A, B)$ and $(Z_{min}(B) < Z_{max}(A) + \epsilon)$ |
|  | $X_{aux}(A, B)$ | $X_{mean}(A) < X_{max}(B)$ and $X_{min}(B) < X_{mean}(A)$ |
|  | $Z_{aux}(A, B)$ | $Z_{mean}(A) < Z_{max}(B)$ and $Z_{min}(B) < Z_{mean}(A)$ |
|  | $h_{aux}(A, B)$ | $closeAbove(A, B)$ or $closeBelow(A, B)$ |
|  | $v_{aux}(A, B)$ | $closeLeftOf(A, B)$ or $closeRightOf(A, B)$ |
|  | $d_{aux}(A, B)$ | $closeInFrontOf(A, B)$ or $closeBehind(A, B)$ |
| spatial | $leftOf(A, B)$ | $X_{mean}(A) < X_{mean}(B))$ |
|  | $above(A, B)$ | $Y_{mean}(A) < Y_{mean}(B)$ |
|  | $inFrontOf(A, B)$ | $Z_{mean}(A) < Z_{mean}(B))$ |
|  | $on(A, B)$ | $closeAbove(A, B)$ and $Z_{aux}(A, B)$ and $X_{aux}(A, B)$ |
|  | $close(A, B)$ | $h_{aux}(A, B)$ or $v_{aux}(A, B)$ or $d_{aux}(A, B)$ |

Table 1: Predicates defining spatial relations between $A$ and $B$. Auxiliary relations define actual spatial relations. The $Y$ axis points downwards, functions $X_{max}, X_{min}, ...$ take appropriate values from the tuple $predicate$, and $\epsilon$ is a 'small' amount. Symmetrical relations such as $rightOf$, $below$, $behind$, etc. can readily be defined in terms of other relations (i.e. $below(A, B) = above(B, A)$).

color [16] (Figure 1 - middle part). Every object hypothesis is therefore represented as an n-tuple: $predicate(instance\_id, image\_id, color, spatial\_loc)$ where $predicate \in \{bag, bed, books, ...\}$, $instance\_id$ is the object's id, $image\_id$ is id of the image containing the object, $color$ is estimated color of the object [16], and $spatial\_loc$ is the object's position in the image. Latter is represented as $(X_{min}, X_{max}, X_{mean}, Y_{min}, Y_{max}, Y_{mean}, Z_{min}, Z_{max}, Z_{mean})$ and defines minimal, maximal, and mean location of the object along $X, Y, Z$ axes. To obtain the coordinates we fit axis parallel cuboids to the cropped 3d objects based on the semantic segmentation. Note that the $X, Y, Z$ coordinate system is aligned with direction of gravity [15]. As shown in Figure 2b, this is a more meaningful representation of the object's coordinates over simple image coordinates. The complete schema will be documented together with the code release.

We realize that the skilled use of spatial relations is a complex task and grounding spatial relations is a research thread on its own (e.g. [17], [18] and [19]). For our purposes, we focus on predefined relations shown in Table 1, while the association of them as well as the object classes are still dealt within the question answering architecture.

**Multi-worlds approach for combining uncertain visual perception and symbolic reasoning**
Up to now we have considered the output of the semantic segmentation as "hard facts", and hence ignored uncertainty in the class labeling. Every such labeling of the segments corresponds to different interpretation of the scene - different perceived world. Drawing on ideas from probabilistic databases [14], we propose a multi-world approach (Figure 1 - lower part) that marginalizes over multiple possible worlds $\mathcal{W}$ - multiple interpretations of a visual scene - derived from the segmentation $\mathcal{S}$. Therefore the posterior over the answer $A$ given question $Q$ and semantic segmentation $S$ of the image marginalizes over the latent worlds $\mathcal{W}$ and logical forms $\mathcal{T}$:

$$P(A \mid Q, S) = \sum_{\mathcal{W}} \sum_{\mathcal{T}} P(A \mid \mathcal{W}, \mathcal{T}) P(\mathcal{W} \mid S) \, P(\mathcal{T} \mid Q) \qquad (2)$$

The semantic segmentation of the image is a set of segments $s_i$ with the associated probabilities $p_{ij}$ over the $C$ object categories $c_j$. More precisely $S = \{(s_1, L_1), (s_2, L_2), ..., (s_k, L_k)\}$ where $L_i = \{(c_j, p_{ij})\}_{j=1}^{C}$, $P(s_i = c_j) = p_{ij}$, and $k$ is the number of segments of given image. Let $\hat{S}_f = \{(s_1, c_{f(1)}), (s_2, c_{f(2)}), ..., (s_k, c_{f(k)})\}$ be an assignment of the categories into segments of the image according to the binding function $f \in \mathcal{F} = \{1, ..., C\}^{\{1, ..., k\}}$. With such notation, for a fixed binding function $f$, a world $\mathcal{W}$ is a set of tuples consistent with $\hat{S}_f$, and define $P(W|S) = \prod_i p_{(i, f(i))}$. Hence we have as many possible worlds as binding functions, that is $C^k$. Eq. 2 becomes quickly intractable for $k$ and $C$ seen in practice, wherefore we use a sampling strategy that draws a finite sample $\vec{\mathcal{W}} = (\mathcal{W}_1, \mathcal{W}_2, ..., \mathcal{W}_N)$ from $P(\cdot|S)$ under an assumption that for each segment $s_i$ every object's category $c_j$ is drawn independently according to $p_{ij}$. A few sampled perceived worlds are shown in Figure 2a.

Regarding the computational efficiency, computing $\sum_{\mathcal{T}} P(A \mid \mathcal{W}_i, \mathcal{T}) P(\mathcal{T} \mid Q)$ can be done independently for every $\mathcal{W}_i$, and therefore in parallel without any need for synchronization. Since for small $N$ the computational costs of summing up computed probabilities is marginal, the overall cost is about the same as single inference modulo parallelism. The presented multi-world approach to question answering on real-world scenes is still an end-to-end architecture that is trained solely on the question-answer pairs.

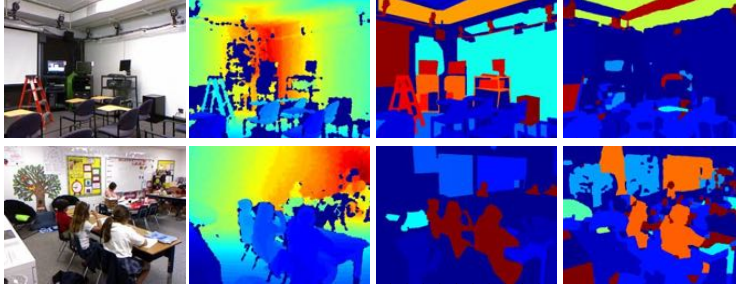

Figure 3: NYU-Depth V2 dataset: image, $Z$ axis, ground truth and predicted semantic segmentations.

| | Description | Template | Example |
|---|---|---|---|
| Individual | counting | How many {object} are in {image_id}? | How many cabinets are in image1? |
| | counting and colors | How many {color} {object} are in {image_id}? | How many gray cabinets are in image1? |
| | room type | Which type of the room is depicted in {image_id}? | Which type of the room is depicted in image1? |
| | superlatives | What is the largest {object} in {image_id}? | What is the largest object in image1? |
| set | counting and colors | How many {color} {object}? | How many black bags? |
| | negations type 1 | Which images do not have {object}? | Which images do not have sofa? |
| | negations type 2 | Which images are not {room_type}? | Which images are not bedroom? |
| | negations type 3 | Which images have {object} but do not have a {object}? | Which images have desk but do not have a lamp? |

Table 2: Synthetic question-answer pairs. The questions can be about individual images or the sets of images.

**Implementation and Scalability**   For worlds containing many facts and spatial relations the induction step becomes computationally demanding as it considers all pairs of the facts (we have about 4 million predicates in the worst case). Therefore we use a batch-based approximation in such situations. Every image induces a set of facts that we call a batch of facts. For every test image, we find $k$ nearest neighbors in the space of training batches with a boolean variant of TF.IDF to measure similarity [20]. This is equivalent to building a training world from $k$ images with most similar content to the perceived world of the test image. We use $k = 3$ and 25 worlds in our experiments. Dataset and the source code can be found in our website [1].

## 4   Experiments

### 4.1   DAtaset for QUestion Answering on Real-world images (DAQUAR)

**Images and Semantic Segmentation**   Our new dataset for question answering is built on top of the NYU-Depth V2 dataset [6]. NYU-Depth V2 contains 1449 RGBD images together with annotated semantic segmentations (Figure 3) where every pixel is labeled into some object class with a confidence score. Originally 894 classes are considered. According to [15], we preprocess the data to obtain canonical views of the scenes and use $X, Y, Z$ coordinates from the depth sensor to define spatial placement of the objects in 3D. To investigate the impact of uncertainty in the visual analysis of the scenes, we also employ computer vision techniques for automatic semantic segmentation. We use a state-of-the-art scene analysis method [15] which maps every pixel into 40 classes: 37 informative object classes as well as 'other structure', 'other furniture' and 'other prop'. We ignore the latter three. We use the same data split as [15]: 795 training and 654 test images. To use our spatial representation on the image content, we fit 3d cuboids to the segmentations.

**New dataset of questions and answers**   In the spirit of a visual turing test, we collect question answer pairs from human annotators for the NYU dataset. In our work, we consider two types of the annotations: synthetic and human. The *synthetic question-answer pairs* are automatically generated question-answer pairs, which are based on the templates shown in Table 2. These templates are then instantiated with facts from the database. To collect 12468 *human question-answer pairs* we ask 5 in-house participants to provide questions and answers. They were instructed to give valid answers that are either basic colors [16], numbers or objects (894 categories) or sets of those. Besides the answers, we don't impose any constraints on the questions. We also don't correct the questions as we believe that the semantic parsers should be robust under the human errors. Finally, we use 6794 training and 5674 test question-answer pairs – about 9 pairs per image on average $(8.63, 8.75)^2$.

The database exhibit some biases showing humans tend to focus on a few prominent objects. For instance we have more than $400$ occurrences of table and chair in the answers. In average the object's category occurs $(14.25, 4)$ times in training set and $(22.48, 5.75)$ times in total. Figure 4 shows example question-answer pairs together with the corresponding image that illustrate some of the challenges captured in this dataset.

**Performance Measure**    While the quality of an answer that the system produces can be measured in terms of accuracy w.r.t. the ground truth (correct/wrong), we propose, inspired from the work on Fuzzy Sets [22], a soft measure based on the WUP score [23], which we call WUPS (WUP Set) score. As the number of classes grows, the semantic boundaries between them are becoming more fuzzy. For example, both concepts 'carton' and 'box' have similar meaning, or 'cup' and 'cup of coffee' are almost indifferent. Therefore we seek a metric that measures the quality of an answer and penalizes naive solutions where the architecture outputs too many or too few answers. Standard Accuracy is defined as: $\frac{1}{N} \sum_{i=1}^{N} \mathbf{1}\{A^i = T^i\} \cdot 100$ where $A^i$, $T^i$ are $i$-th answer and ground-truth respectively. Since both the answers may include more than one object, it is beneficial to represent them as sets of the objects $T = \{t_1, t_2, ...\}$. From this point of view we have for every $i \in \{1, 2, ..., N\}$:

$$\mathbf{1}\{A^i = T^i\} = \mathbf{1}\{A^i \subseteq T^i \cap T^i \subseteq A^i\} = \min\{\mathbf{1}\{A^i \subseteq T^i\}, \ \mathbf{1}\{T^i \subseteq A^i\}\} \quad (3)$$

$$= \min\{\prod_{a \in A^i} \mathbf{1}\{a \in T^i\}, \ \prod_{t \in T^i} \mathbf{1}\{t \in A^i\}\} \approx \min\{\prod_{a \in A^i} \mu(a \in T^i), \ \prod_{t \in T^i} \mu(t \in A^i)\} \quad (4)$$

We use a soft equivalent of the intersection operator in Eq. 3, and a set membership measure $\mu$, with properties $\mu(x \in X) = 1$ if $x \in X$, $\mu(x \in X) = \max_{y \in X} \mu(x = y)$ and $\mu(x = y) \in [0, 1]$, in Eq. 4 with equality whenever $\mu = \mathbf{1}$. For $\mu$ we use a variant of Wu-Palmer similarity [23, 24]. $\mathrm{WUP}(a, b)$ calculates similarity based on the depth of two words $a$ and $b$ in the taxonomy[25, 26], and define the WUPS score:

$$\mathrm{WUPS}(A, T) = \frac{1}{N} \sum_{i=1}^{N} \min\{\prod_{a \in A^i} \max_{t \in T^i} \mathrm{WUP}(a, t), \ \prod_{t \in T^i} \max_{a \in A^i} \mathrm{WUP}(a, t)\} \cdot 100 \quad (5)$$

Empirically, we have found that in our task a WUP score of around $0.9$ is required for precise answers. Therefore we have implemented down-weighting $\mathrm{WUP}(a, b)$ by one order of magnitude $(0.1 \cdot \mathrm{WUP})$ whenever $\mathrm{WUP}(a, b) < t$ for a threshold $t$. We plot a curve over thresholds $t$ ranging from $0$ to $1$ (Figure 5). Since "WUPS at 0" refers to the most 'forgivable' measure without any down-weighting and "WUPS at 1.0" corresponds to plain accuracy. Figure 5 benchmarks architectures by requiring answers with precision ranging from low to high. Here we show some examples of the pure WUP score to give intuitions about the range: WUP(curtain, blinds) $= 0.94$, WUP(carton, box) $= 0.94$, WUP(stove, fire extinguisher) $= 0.82$.

## 4.2   Quantitative results

We perform a series of experiments to highlight particular challenges like uncertain segmentations, unknown true logical forms, some linguistic phenomena as well as show the advantages of our proposed multi-world approach. In particular, we distinguish between experiments on synthetic question-answer pairs (**SynthQA**) based on templates and those collected by annotators (**HumanQA**), automatic scene segmentation (**AutoSeg**) with a computer vision algorithm [15] and human segmentations (**HumanSeg**) based on the ground-truth annotations in the NYU dataset as well as single world (**single**) and multi-world (**multi**) approaches.

### 4.2.1   Synthetic question-answer pairs (SynthQA)

**Based on human segmentations (HumanSeg, 37 classes)** (1st and 2nd rows in Table 3) uses automatically generated questions (we use templates shown in Table 2) and human segmentations. We have generated $20$ training and $40$ test question-answer pairs per template category, in total $140$ training and $280$ test pairs (as an exception negations type 1 and 2 have $10$ training and $20$ test examples each). This experiment shows how the architecture generalizes across similar type of questions provided that we have human annotation of the image segments. We have further removed negations of type 3 in the experiments as they have turned out to be particularly computationally demanding. Performance increases hereby from $56\%$ to $59.9\%$ with about $80\%$ training Accuracy. Since some incorrect derivations give correct answers, the semantic parser learns wrong associations. Other difficulties stem from the limited training data and unseen object categories during training.

**Based on automatic segmentations (AutoSeg, 37 classes, single)** (3rd row in Table 3) tests the architecture based on uncertain facts obtained from automatic semantic segmentation [15] where the

most likely object labels are used to create a single world. Here, we are experiencing a severe drop in performance from 59.9% to 11.25% by switching from human to automatic segmentation. Note that there are only 37 classes available to us. This result suggests that the vision part is a serious bottleneck of the whole architecture.

**Based on automatic segmentations using multi-world approach (AutoSeg, 37 classes, multi)** (4th row in Table 3) shows the benefits of using our multiple worlds approach to predict the answer. Here we recover part of the lost performance by an explicit treatment of the uncertainty in the segmentations. Performance increases from 11.25% to 13.75%.

### 4.3 Human question-answer pairs (HumanQA)

**Based on human segmentations 894 classes (HumanSeg, 894 classes)** (1st row in Table 4) switching to human generated question-answer pairs. The increase in complexity is twofold. First, the human annotations exhibit more variations than the synthetic approach based on templates. Second, the questions are typically longer and include more spatially related objects. Figure 4 shows a few samples from our dataset that highlights challenges including complex and nested spatial reference and use of reference frames. We yield an accuracy of 7.86% in this scenario. As argued above, we also evaluate the experiments on the human data under the softer WUPS scores given different thresholds (Table 4 and Figure 5). In order to put these numbers in perspective, we also show performance numbers for two simple methods: predicting the most popular answer yields 4.4% Accuracy, and our untrained architecture gives 0.18% and 1.3% Accuracy and WUPS (at 0.9).

**Based on human segmentations 37 classes (HumanSeg, 37 classes)** (2nd row in Table 4) uses human segmentation and question-answer pairs. Since only 37 classes are supported by our automatic segmentation algorithm, we run on a subset of the whole dataset. We choose the 25 test images yielding a total of 286 question answer pairs for the following experiments. This yields 12.47% and 15.89% Accuracy and WUPS at 0.9 respectively.

**Based on automatic segmentations (AutoSeg, 37 classes)** (3rd row in Table 4) Switching from the human segmentations to the automatic yields again a drop from 12.47% to 9.69% in Accuracy and we observe a similar trend for the whole spectrum of the WUPS scores.

**Based on automatic segmentations using multi-world approach (AutoSeg, 37 classes, multi)** (4th row in Table 4) Similar to the synthetic experiments our proposed multi-world approach yields an improvement across all the measure that we investigate.

**Human baseline** (5th and 6th rows in Table 4 for 894 and 37 classes) shows human predictions on our dataset. We ask independent annotators to provide answers on the questions we have collected. They are instructed to answer with a number, basic colors [16], or objects (from 37 or 894 categories) or set of those. This performance gives a practical upper bound for the question-answering algorithms with an accuracy of 60.27% for the 37 class case and 50.20% for the 894 class case. We also ask to compare the answers of the AutoSeg single world approach with HumanSeg single world and AutoSeg multi-worlds methods. We use a two-sided binomial test to check if difference in preferences is statistically significant. As a result AutoSeg single world is the least preferred method with the p-value below 0.01 in both cases. Hence the human preferences are aligned with our accuracy measures in Table 4.

### 4.4 Qualitative results

We choose examples in Fig. 6 to illustrate different failure cases - including last example where all methods fail. Since our multi-world approach generates different sets of facts about the perceived worlds, we observe a trend towards a better representation of high level concepts like 'counting' (leftmost the figure) as well as language associations. A substantial part of incorrect answers is attributed to missing segments, e.g. no pillow detection in third example in Fig. 6.

## 5 Summary

We propose a system and a dataset for question answering about real-world scenes that is reminiscent of a visual turing test. Despite the complexity in uncertain visual perception, language understanding and program induction, our results indicate promising progress in this direction. We bring ideas together from automatic scene analysis, semantic parsing with symbolic reasoning, and combine them under a multi-world approach. As we have mature techniques in machine learning, computer vision, natural language processing and deduction at our disposal, it seems timely to bring these disciplines together on this open challenge.

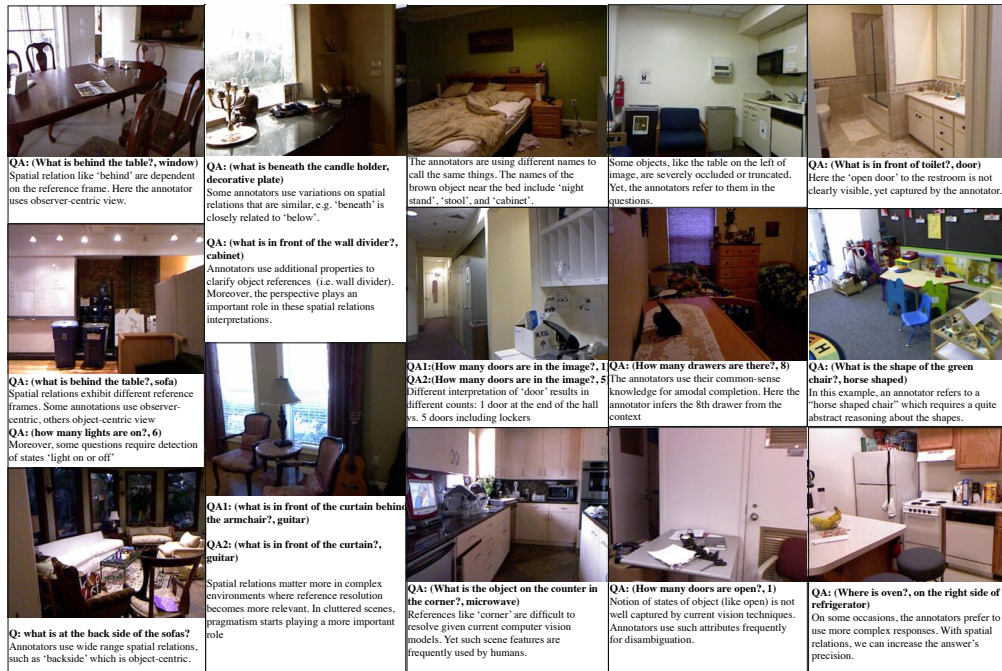

Figure 4: Examples of human generated question-answer pairs illustrating the associated challenges. In the descriptions we use following notation: 'A' - answer, 'Q' - question, 'QA' - question-answer pair. Last two examples (bottom-right column) are from the extended dataset not used in our experiments.

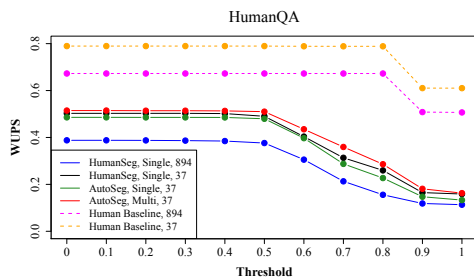

Figure 5: WUPS scores for different thresholds.

| synthetic question-answer pairs (SynthQA) | | | |
|---|---|---|---|
| Segmentation | World(s) | # classes | Accuracy |
| HumanSeg | Single with Neg. 3 | 37 | 56.0% |
| HumanSeg | Single | 37 | 59.5% |
| AutoSeg | Single | 37 | 11.25% |
| AutoSeg | Multi | 37 | 13.75% |

Table 3: Accuracy results for the experiments with synthetic question-answer pairs.

| Human question-answer pairs (HumanQA) | | | | | |
|---|---|---|---|---|---|
| Segmentation | World(s) | #classes | Accuracy | WUPS at 0.9 | WUPS at 0 |
| HumanSeg | Single | 894 | 7.86% | 11.86% | 38.79% |
| HumanSeg | Single | 37 | 12.47% | 16.49% | 50.28% |
| AutoSeg | Single | 37 | 9.69% | 14.73% | 48.57% |
| AutoSeg | Multi | 37 | 12.73% | 18.10% | 51.47% |
| Human Baseline | | 894 | 50.20% | 50.82% | 67.27% |
| Human Baseline | | 37 | 60.27% | 61.04% | 78.96% |

Table 4: Accuracy and WUPS scores for the experiments with human question-answer pairs. We show WUPS scores at two opposite sides of the WUPS spectrum.

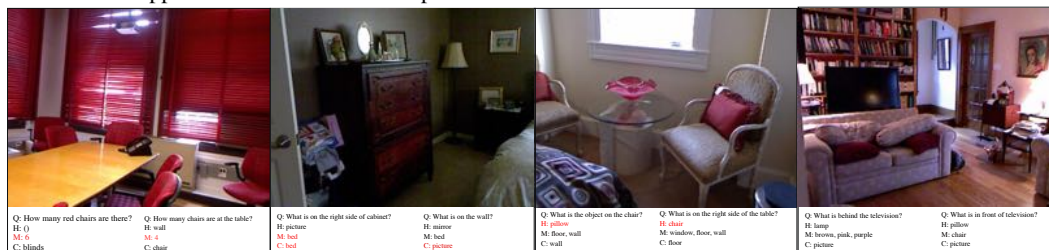

Figure 6: Questions and predicted answers. Notation: 'Q' - question, 'H' - architecture based on human segmentation, 'M' - architecture with multiple worlds, 'C' - most confident architecture, '()' - no answer. Red color denotes correct answer.

## Footnotes

[1] https://www.d2.mpi-inf.mpg.de/visual-turing-challenge

[2] Our notation $(x, y)$ denotes mean $x$ and trimean $y$. We use Tukey's trimean $\frac{1}{4}(Q_1 + 2Q_2 + Q_3)$, where $Q_j$ denotes the $j$-th quartile [21]. This measure combines the benefits of both median (robustness to the extremes) and empirical mean (attention to the hinge values).

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
