[Reviews · NeurIPS 2014]

Submitted by Assigned_Reviewer_14

The authors present a method for question answering about real world scenes - given as input a real world image and a question regarding objects in this image their system answers this question. For the question-answering engine the authors have generated a novel dataset with more than 12k question-answer pairs.

- The paper is well written and I believe this combination of NLP and CV to perceive the world will be of great interest to the NIPS community.
- The dataset that the authors have presented would be of great value to the community.

- From the algorithmic aspect the contribution of the authors is in proposing the use of computer vision for semantic segmentation and the multi-world approach rather than the human segmented single-world approach as used in [5]. The authors show an improved performance when using the multi-world approach but it didn't fully convinced me as for its quality since the accuracy (and WUPS) is pretty low either way. I would like to see more evidence and understanding of the importance and contribution of the multi-world approach. In addition, I am also don't agree with the assumption (stated in line 215) that for each segment every object's category is drawn independently. I would consider to use the best-diverse-M segmentation approach as in Yadollahpour et al, Discriminative Re-ranking of Diverse Segmentations.
- The overall performance is not very impressing, though I believe it is a good starting point.
Summary: Overall I think the paper will be of great interest to the NIPS community and the presentation of the QA dataset will help to push this research area forward. My concerns are regarding the algorithmic novelty as it seems very close to the work presented in [5] and the not so high performance.

Submitted by Assigned_Reviewer_17

This paper created a question-answering dataset annotated on a set of NYU-depth images with gold segmentation. The paper also adapted Liang et al. 2011 to learn semantic parser from QA pairs, using the segmentation annotation and hand-crafted denotation for spatial predicates to derive answer from parse during learning.

This paper is an interesting exploration of grounded learning in image QA, which should be encouraged. Unfortunately, the paper seemed rather rush and was not very clear in many important details, making it difficult to assess the true significance of the proposed work.

First, it is not entirely clear what's the ontology commitment and how answer is derived. In Liang et al. 2011, the database schema provides the former, and the latter comes from relational algebra. Here, the details are only glossed over. It's unclear what are the set of predicates (and how many), how aggregation and superlative are handled, etc. It will be helpful to provide an end-to-end example.

In terms of semantic parsing, it's unclear what features were used in the log-linear model. The paper describes an ad hoc procedure to find k nearest neighbors for a test image from training images. It's unclear what this was for and how it was used. Were these used in training or test? Why picks k=3 and 25 worlds? The paper suggested that these procedures stemmed from scalability problem. What's the complexity in train/test? How difficult could the proposed approach generalizes to bigger datasets, larger images, and longer questions?

It's also unclear how the QA pairs were derived. Who are the annotators? What's the annotation guideline? How long are these questions? What is the distribution of question types? Did the authors evaluate answer quality and agreement?

The evaluation is quite confusing. The paper said WUP is down-weighted if it's less than 0.9, and then provide a few WUPs, with WUP(stove, fire extinguisher)=0.82. What's the WUP in use in (6)? Should it be 0.82 or 0.082? The down-weighting also appears rather arbitrary. I also have a hard time guessing what "WUP at 0" signifies in Table 4.

Regarding the long-term direction, it's worrisome, though not surprising, to see the dramatic drop of performance when manual segmentation is replaced with automatic one in synthetic QA. (For human QA, the drop is smaller, though the accuracy is very low to begin with.) It'll also be useful to discuss scalability in light of long-term feasibility.
Summary: This paper is an interesting exploration of grounded learning in image QA, which should be encouraged. Unfortunately, the paper seemed rather rush and was not very clear in many important details, making it difficult to assess the true significance of the proposed work.

Submitted by Assigned_Reviewer_22

This paper describes a dataset and system for answering natural language questions about a depth/RGB image, which uses multiple segmentations to capture uncertainty in the scene understanding when computing answers.

Quality
=======
This is an extremely difficult task which requires understanding natural language forms of queries, segmenting and categorising the depth image, understanding the scene layout in three dimensions (including some understanding of support) and bringing all these pieces together. The state of the art in each part individually is not currently close to having good accuracy - one concern is that it is premature to try and tie these systems together to solve an even harder problem. The worry is that errors will just accumulate from system to system - although it certainly helps that some uncertainty in the segmentation is communicated though samples (multiple worlds). This concern seems borne out by the fact that the overall system is giving results only slightly better than always picking the same label all the time (7.86% vs 4.4%). I wonder how much better the system is doing than a colour/word histogram-based straw man? With just 4 examples (Fig 6) It's hard to tell the kinds of mistakes the system is making and which piece of the system is the cause, but the fact that it is getting the type wrong (reporting a colour in answer to "what is behind the television?") suggests that there's room for improvement in the question-answering part alone.

The WUPS evaluation metric seems to have some issues - the fact that humans doing so badly (61% at 0.9) seems more likely to be an issue with the metric than with the humans. I'm not quite sure what the sentence "Despite our instructions the answers are subject to... human judgement" is implying! It might be interesting to explore an evaluation metric that says whether a human thinks the automatically-given answer is correct or not - or at least use this to validate the WUPS metric.

Clarity
=======
Overall acceptably clear, but with a lot of missing details which require reading [1] or are just not present e.g. the definition of the 'sampling strategy' used to produce segmentation samples. The figures could use refinement e.g. the segmentation images lack a key, the text in Fig 6 is hard to read. The authors should also say how the examples in Fig 6 were selected (at random? are these typical?). The introduction could use some references for the methods being referred to as could the second sentence of section 3.

Originality & significance
==========================
The question/answer dataset which augments the existing NYU depth/RGB image dataset is a useful contribution. Given the extreme difficulty of the task and the issues with the individual components, the contribution of combining them is mainly to get a straw man accuracy number.

Typos:
Pg 2 "[4] objects" - words missing
Bottom of pg 4 "words"->"worlds"
Refs: many capitalization issues.
Summary: This paper tackles an extremely hard problem by creating a new dataset and then tying together existing systems to attack the problem. It is great that the authors are tackling such a challenging problem - the only concern is that it feels like they bit off more than they could chew, given the relatively poor results. The presentation could use some refinement and there are some worries about the evaluation metric, but it is nonetheless an impressive achievement to get the overall system to work at all given the challenges with the individual components.
Author Feedback
Author rebuttal: We thank all the reviewers for their comments and suggestions. We will improve the presentation of the paper accordingly. The reviewers acknowledge that our work is "an impressive achievement to get the overall system to work" on an"extremely hard problem" of a question-answering task. To achieve our goals we propose a
"novel dataset with more than 12k question-answer pairs" that "would be of great value to the community".
To deal with the uncertainty that comes from perception, we propose and experimentally validate the multi-world approach that marginalizes over multiple uncertain interpretations of the world. Finally, our inference unites both the latent language representation and multiple worlds in a probabilistic graphical model (Fig. 1).

Although the scene and natural language understanding tasks are not solved yet, we strongly believe that both technologies matured to a level where we can work towards addressing a question-answering challenge that relates to the long term goal of a visual turing test. We build the dataset, and based on the recent achievements in both communities, we believe that the time is right to make a push forward in this direction.
To stimulate research in this direction we will make the dataset and our code available - as promised in the paper.

AR14:
In contrast to [5], we deal with a larger and more diverse collection of images with more objects, attributes, complex spatial configurations and a richer class of questions (i.e. superlatives and negations; see Table 2 for all types).
This imposes additional challenges for the question-answering engines such as scalability of the semantic parser,
good scene representation, dealing with uncertainty in the language and perception, efficient inference and spatial reasoning. We also propose alternative performance measure and present a much broader experimental studies.

A single world approach cannot answer on the questions about the objects that are not in the knowledge base. By sampling, our multi-world approach reduces these problems that stem from uncertain and ambiguous perception.

Independence between the objects in our sampling scheme is definitely a simplifying assumption. We are grateful for the suggestions about introducing dependencies in the sampling process which will be explored in future work.

AR17:
We use the same language representation, semantic parser and relation algebra as Liang [1] with the same features (section 3 in [1]) to encode semantic trees. The complete schema will be documented together with the code release.

We use 894 predicates (#classes 894) and 37 predicates (#classes 37)
representing the whole set and a subset of objects in NYU_Depth.
Every predicate represents an object (schema in lines 177-185) from either manual or automatic segmentations.
Our lexical triggers include about 20 function words, and a set of predicates for POS tags (NN,NNS, JJ); it corresponds to DCS with L in [1]. As [1] we use the Prolog inference engine to derive answers.
DCS [1] is our formal language. It handles counting, superlatives, spatial relations, attributes and negations (Table 2).

[1] doesn’t scale well wrt. spatial relations and negations. Apart from that we didn’t notice other scalability issues. As a solution, we propose a KNN approximation for training.
We chose K and number of worlds based on the computational efficiency. We will include an analysis of the speed accuracy tradeoff for different K in the final version.

The program induction - the core element of [1] - searches over the exponential space of all possible formalisms.
In practice we use a beam search (also suggested in [1]) to control the complexity growth.

Regarding annotations: We ask 5 participants to provide questions and answers.
The annotators were instructed to provide either colors, numbers, objects (894 categories) or sets of those in the answers (lines 269-272). We didn't impose any further constraints and only fixed spelling errors.
The agreement of the answers can be approximated by the Human Baseline where we asked independent participants to answer the questions (lines 352-357).
We are steadily growing the dataset in order to account for ambiguities in the answers.
The longest question has 30 words, shortest 4 words, the average is 10.5 words with variance 5.5.

WUP is Wu-Palmer similarity with a few examples (lines 291-305). Motivated by inspecting the scores, we use a threshold to down weight WUP in the WUPS scores.
To avoid arbitrary choices we suggest to compare the methods across all thresholds between 0 and 1 (Fig. 5).
Table 4 shows results at two extremes (WUPS at 0 and 0.9). WUPS at 0 means ‘no thresholding’ and therefore no down-weighting.

AR22:
The imperfect scores in the Human Baselines indicate ambiguities in the answers and show the difficulty of the task.

Clarification on: "Despite our instructions the answers are subject to typical human errors and/or human judgement.":
The annotators were instructed to use only the objects from our vocabulary (lines 352-357). A few answers diverge from these instructions and use different words. In addition, answers vary naturally due to ambiguity in the interpretation of the question.

Our goal is to make a public benchmark and hence we are mostly interested in the automatic evaluations. However, we agree to include a study that shows human judgement of the results on a subset.

Many worlds are created by sampling the object labels for every segmentation based on the confidence scores of the detections (lines 201-215).

Examples in Fig.6 were randomly chosen to illustrate different failure cases - including last example where all methods fail.

=
[1] P. Liang, et. al. "Learning dependency-based compositional semantics"
[5] J. Krishnamurthy, et. al. "Jointly Learning to Parse and Perceive: Connecting Natural Language to the Physical World"